# What Are the Current Approaches to Optimising Antimicrobial Dosing in the Intensive Care Unit?

**DOI:** 10.3390/pharmaceutics12070638

**Published:** 2020-07-07

**Authors:** Ming G. Chai, Menino O. Cotta, Mohd H. Abdul-Aziz, Jason A. Roberts

**Affiliations:** 1University of Queensland Centre for Clinical Research (UQCCR), Faculty of Medicine, The University of Queensland, Brisbane 4006, Australia; g.chai@uq.net.au (M.G.C.); m.o.cotta@uq.edu.au (M.O.C.); h.abdulaziz@uq.edu.au (M.H.A.-A.); 2Centre for Translational Anti-infective Pharmacodynamics, School of Pharmacy, The University of Queensland, Woollongabba 4102, Australia; 3Departments of Pharmacy and Intensive Care, Royal Brisbane and Women’s Hospital, Brisbane 4006, Australia; 4Division of Anaesthesiology Critical Care Emergency and Pain Medicine, Nimes University Hospital, University of Montpellier, 30021 Nimes, France

**Keywords:** nomogram, software, antimicrobials, critical illness, pharmacokinetics, Bayesian forecasting, sepsis, artificial intelligence

## Abstract

Antimicrobial dosing in the intensive care unit (ICU) can be problematic due to various challenges including unique physiological changes observed in critically ill patients and the presence of pathogens with reduced susceptibility. These challenges result in reduced likelihood of standard antimicrobial dosing regimens achieving target exposures associated with optimal patient outcomes. Therefore, the aim of this review is to explore the various methods for optimisation of antimicrobial dosing in ICU patients. Dosing nomograms developed from pharmacokinetic/statistical models and therapeutic drug monitoring are commonly used. However, recent advances in mathematical and statistical modelling have resulted in the development of novel dosing software that utilise Bayesian forecasting and/or artificial intelligence. These programs utilise therapeutic drug monitoring results to further personalise antimicrobial therapy based on each patient’s clinical characteristics. Studies quantifying the clinical and cost benefits associated with dosing software are required before widespread use as a point-of-care system can be justified.

## 1. Introduction

Sepsis is a leading source of morbidity and mortality among critically ill patients in the intensive care unit (ICU) [1,2,3]. In addition to selecting the most appropriate antimicrobial agent, there is increasing awareness of how dosing strategies employed can have profound implications on the success of therapy in this patient group [4,5]. Suboptimal antimicrobial dosing reduces the likelihood of achieving pharmacokinetic-pharmacodynamic (PK-PD) targets needed for therapeutic success. With an increasing appreciation of the importance of achieving these PK-PD targets, regulatory bodies are progressively incorporating the use of PK-PD studies through pharmacometrics as part of their regulatory assessment of new antimicrobials and their licensed doses [6]. Critically ill patients, in particular, are at an increased risk of treatment failure due to the unique physiological changes commonly seen in these patients and the interventions they are exposed to that often result in alterations to antimicrobial PK and achievement of PK-PD targets.

### 1.1. Altered Pharmacokinetics

Traditionally, drug dosing takes a “one-size-fits-all” approach whereby pharmacokinetic (PK) data is used to define dosing regimens to be used in indications for which the drug is licensed. This PK data predicts the likely drug exposure that can be obtained from a chosen antimicrobial dosing regimen [7]. As these studies are typically conducted in healthy volunteers and not severely unwell patients, extrapolating these dosing recommendations to other patient groups does not account for altered PK that is often observed in this such populations, especially in the critically ill [8].

A common finding in critically ill patients with sepsis is the presence of large fluid shifts (or the third-spacing phenomenon) into the interstitial space, which can alter antimicrobial exposure by increasing their volume of distribution. Through movement of drug into this additional compartment, there is less drug available in plasma and potentially at the site of infection [9], reducing the likelihood of optimal drug exposure and patient outcomes [10]. Hydrophilic antimicrobials, such as aminoglycosides and beta-lactams, are more affected by this fluid redistribution than lipophilic antimicrobials [11]. Other physiological changes seen in sepsis include the presence of organ dysfunction. Given that the hepatic [12,13,14] and renal systems [15,16,17,18] are responsible for metabolism and excretion of many antimicrobial agents, derangements in function are likely to result in drug accumulation. If antimicrobial dosing is not adapted to organ dysfunction, drug-related toxicity from high drug exposures may result [19,20].

Alternatively, increased cardiac output from sepsis and inotrope/vasopressor use have been shown to increase renal perfusion and induce augmented renal clearance (ARC, defined as a glomerular filtration rate above 130 mL/min/1.73 m^2^ [21]). ARC can dramatically increase the clearance of renally-cleared hydrophilic antimicrobials [22,23]. Reduction in plasma albumin concentrations is common in critically ill patients which can increase clearance of highly protein-bound antimicrobials (and reduced concentrations), for example, ceftriaxone and teicoplanin, as there is an increase in free drug available for clearance [24]. These collective physiological changes have been shown to reduce the likelihood of achieving exposure targets needed for antimicrobial efficacy and potentially increasing the risk of treatment failure [10,19,25,26,27,28].

An additional level of complexity to antimicrobial dosing in critically ill patients stems from the treatments used to support failing organs. Extracorporeal interventions such as extracorporeal membrane oxygenation (ECMO) and renal replacement therapy (RRT), although lifesaving, can profoundly alter antimicrobial exposure in these patients. Adult patients exposed to ECMO can have further increases in the volume of distribution of highly protein-bound and/or lipophilic antimicrobials through sequestration onto the ECMO circuits [29,30,31]. Compared to non-dialysed patients with renal failure, patients receiving RRT may exhibit significant clearances of hydrophilic antimicrobials through losses in the ultrafiltrate and dialysate [32]. With both interventions, the impact of drug clearance or distribution is affected by a wide variety of extracorporeal parameters including the type of material used for the oxygenator or filter and their surface areas, blood and effluent flow rates, configuration or modality of the intervention used as well as replacement fluid settings, especially for RRT. Hence, these interventions make it difficult to predict resulting antimicrobial exposure given the large variations in extracorporeal parameters used in clinical settings. If ECMO and/or RRT are unaccounted for with the dosing regimen selected, this may lead to treatment failure or drug toxicity. Collectively, optimising antimicrobial therapy in this patient group is critical to ensure positive patient outcomes as these patients are often the sickest cohort in the ICU [33,34,35,36,37].

### 1.2. Pharmacodynamic Considerations

Another major factor that impacts on treatment success is the pharmacodynamics (PD) of an antimicrobial. PK-PD describes the optimum unbound exposure to an antimicrobial agent that is needed for treatment success and is influenced by the primary PD parameter, the minimum inhibitory concentration (MIC) [38]. The MIC is an in vitro measurement that describes the susceptibility of an organism to an antimicrobial agent and hence affects the exposure required for antimicrobial efficacy. Examples of PK-PD indices used to measure antimicrobial efficacy include the ratio of the area under the curve of the unbound drug (AUC) to the MIC (*f*AUC_0–24/_MIC; where *f* denotes free, or unbound exposure), the ratio of the maximal unbound drug concentration to the MIC (*f*C_max_/MIC) and the cumulative percentage of a dosing interval that the antimicrobial concentration exceeds the MIC (%*f*T > MIC) [39]. Although not all PK-PD indices in studies are described according to the unbound concentration (only total concentration may be measured), it is the unbound concentration of drug that is responsible for antimicrobial effect. Table 1 shows the PK-PD indices used to describe microbial kill characteristics of commonly used antimicrobial agents.

Pathogens isolated in critically ill patients often exhibit higher MICs compared to those isolated among ward-based patients [55,56,57,58,59,60,61]. This may mean that higher antimicrobial exposures are needed to attain PK-PD targets associated with optimal clinical outcome. Together with the altered PK observed in these patients, these scenarios are very challenging when faced by ICU clinicians [62]. Furthermore, sub-optimal therapy carries an increased risk of developing antimicrobial-resistant pathogens which has negative consequences in patients both in and outside of the ICU [63].

The pathogen’s MIC can influence the antimicrobial exposure required to achieve a PK-PD target as it acts as the denominator in each index. The MIC should be interpreted in the context of the microbiological susceptibility testing method used (such broth microdilution versus E-test), the pathogen identified and its wild-type distribution. However, it is important to note that the measurement process of MICs is susceptible to laboratory assay and microbiological sample variations [38]. This may inadvertently result in the MIC being reported incorrectly by one to two dilutions and this can affect the achievement of appropriate PD targets, although the clinical outcome implications of MIC measurement error remain uncertain.

Collectively, the dynamic interplay between PK and PD demonstrates how antimicrobial exposure impacts on the critical care patient (Figure 1). Clinicians should adopt a PK-PD based approach to dosing in order to avoid the risks of sub-optimal exposure associated with using standard antimicrobial doses.

The purpose of this review is to describe tools currently available to assist clinicians to achieve therapeutic PK-PD targets by individualising dosing. These include use of dosing nomograms, therapeutic drug monitoring, and dosing software where each tool can function individually or in conjunction with each other to optimise the dosing of antimicrobials (see Figure 2).

## 2. Dosing Nomograms

A simple but systematic approach to guide dosing of antimicrobials is the use of dosing nomograms. This is one of the most common approaches taken in non-critical care settings whereby doses are based on patient characteristics (typically renal function or weight) or antimicrobial plasma concentrations, if available [64]. Nomograms are developed from PK studies or statistical analyses (such as multiple linear regression models) in the population of interest and seek to describe the dose-concentration relationship of an antimicrobial. From these data, a model is generated that then will underpin the dosing recommendations of the nomogram using relevant patient characteristics [65]. Based on patient characteristics, an individualised starting dose likely to achieve the nominated PK-PD targets can be generated from the dosing nomogram. Additionally, some nomograms have the ability to generate subsequent dosage adjustment recommendations using resulting antimicrobial plasma concentrations if available [66].

Compared to clinician-guided dosing alone, use of dosing nomograms in the ICU has shown some promise. For example, an improvement in target attainment of vancomycin plasma concentrations (defined in this study as a trough concentration between 20–30 mg/L) was seen within the first day of treatment with 84% of patients achieving therapeutic concentrations when a dosing nomogram was used compared to 51% of patients when dosed empirically by clinicians [67]. Other nomograms for antimicrobial drugs in the ICU population have also been described for vancomycin [68,69,70,71], gentamicin [72] and meropenem [73].

A limitation to using dosing nomograms in the ICU is its ability to consider only one or two patient characteristics at a time, with additional inputs complicating the feasibility of a nomogram and impair usability. This reduces the clinician’s ability to include more data such as pathogen-specific information to further individualise the dosing recommendations of the nomogram. The dosing recommendations from these nomograms are typically aimed at achieving pre-defined PK targets [66,74] which may not be universally applicable to all ICU patients, given that some patients may be infected with pathogens with higher MICs.

In spite of the limitations described, nomograms may still be a useful tool for improving antimicrobial dosing to help achieve PK-PD targets when compared to empiric dosing regimens selected by clinicians alone. The use of nomograms generally does not require significant changes to pre-existing infrastructure (such as new assays for measuring antimicrobial concentrations) beyond clinician education which may make implementation easier, even in ICUs with more limited resources such as those in low and middle income countries.

## 3. Therapeutic Drug Monitoring in the Intensive Care Unit (ICU)

The use of therapeutic drug monitoring (TDM) represents one of the earliest forms of personalising antimicrobial therapy/dosing [75]. TDM was traditionally used to ensure that patients receiving antimicrobials with a narrow therapeutic index were not exposed to toxic exposures associated with serious side effects. Examples of antimicrobials that were targeted for this purpose include aminoglycosides and glycopeptides where the risk of nephro- and oto-toxicity is higher than for other antimicrobials [76]. With an increasing appreciation of how achieving PK-PD targets increases the likelihood of treatment success, TDM has now expanded to include optimising antimicrobial exposure to increase the likelihood of clinical and microbiological cure, as well as, minimise drug toxicity.

At its simplest, TDM entails obtaining a plasma drug concentration during a course of antimicrobial therapy. This concentration is interpreted by the clinician to be either therapeutic or not therapeutic, in which case the clinician will adjust the dosing regimen by a magnitude that they anticipate will achieve a predefined PK-PD target. Although use of TDM has been shown to increase the proportion of patients who achieve PK-PD targets of antimicrobials compared to empiric dosing by clinicians alone [77], it is prone to significant inter-clinician variability in dosing recommendation selected [78]. This may reduce the consistency of achieving these PK-PD targets when dosing is led by different clinicians. Furthermore, variability between countries and healthcare networks with respect to their TDM practices and availability may increase variability in dosing recommendations across institutions [79,80]. Importantly, dosing changes may be more challenging in antimicrobials where exposures can be influenced by multiple factors such as renal function and plasma protein binding. To circumvent these hurdles, TDM results can be integrated with other tools such as dosing nomograms and/or software to help improve the likelihood that revised doses will achieve predefined PK-PD targets.

There are several considerations clinicians should be aware of when employing TDM, including the availability of appropriate assays needed to generate the TDM results [81]. Not all ICUs have access to these assays which may explain the variability across ICUs in antimicrobials that are actively monitored using this process [79]. Furthermore, the majority of data and suggested reference ranges are based on concentrations obtained from blood (or plasma). These antimicrobial concentrations do not always reflect concentrations at the site of infection (e.g., pulmonary epithelial lining fluid for pneumonia or cerebrospinal fluid for meningitis) [82]. Although preliminary studies have attempted to generate models that predict the antimicrobial concentrations at the site of infection from blood plasma concentrations, their predictive performance has been generally poor [83,84]. For an antimicrobial to reach the site of infection, it must partition out of circulation, diffuse through the interstitial fluid and finally pass into the tissue of interest [85]. Depending on the location of this tissue, a wide variety of factors can influence penetration into the infective site. Examples include drug-specific parameters such as degree of protein binding and physicochemical properties of the drug as well as tissue properties which may include the presence of occluding tissue borders (such as the blood-brain barrier) or membrane transporters (which can enhance or hinder the transport of antimicrobials to and from the site of infection) [85]. In these clinical scenarios, plasma concentrations act as surrogate markers as sampling at the clinical site of infection through procedures such as bronchoalveolar lavage in pneumonia or lumbar puncture in central nervous system (CNS) infections can often be impractical or too invasive to be conducted on a routine basis.

Given this, clinicians may choose to aim for higher plasma concentrations to create a larger concentration gradient with tissues to drive the antimicrobial into the site of infection if a drug is known to have variable penetration. Vancomycin for treating CNS infections is a common example whereby in practice trough concentrations are targeted up to 20–25 mg/L (as opposed to 15–20 mg/L for other infections) due to reduced penetration across the blood-brain barrier which may also be variable depending on the degree of meningeal inflammation [86]. Further studies are needed in this area to better define plasma exposure targets of antimicrobials that best correlate with optimal tissue concentrations and patient outcomes.

Additionally, as only the unbound drug is pharmacologically active to exert a therapeutic effect, highly protein-bound antimicrobial agents (usually clinically significant when more than 70% of the drug is protein bound [87]) warrant additional considerations given the reduction in protein concentrations commonly seen in critically ill patients. Plasma concentrations are typically reported as ‘total’ drug concentrations as measuring only the unbound concentration is more onerous and expensive [88]. However, in ICU patients, measuring total plasma concentrations may not provide an accurate reflection of the unbound concentration needed to achieve PK-PD targets. Previous studies have demonstrated that highly protein-bound antimicrobials such as ceftriaxone and flucloxacillin have poor correlations between their total and unbound concentrations, making it difficult to predict if appropriate target unbound exposures have been achieved when only total concentrations are measured [89]. In patients prescribed drugs with high and/or variable protein binding, adjusting antimicrobial regimens based on total concentrations may result in inaccurate PK-PD target attainment and unbound concentrations should be measured if available. Assays validated to measure the unbound concentration of various antimicrobials have been developed [88].

## 4. Dosing Software

With advances in the ability of computers to perform complex mathematical modelling and statistical analysis, clinicians have access to integrate PK models and/or pharmaco-statistical outputs that have been embedded into dosing software to assist with drug dosing. Depending on the model underpinning the program, dosing software can be broadly divided into a system that utilises (a) linear regression models, (b) population PK models, and/or (c) models that incorporate Bayesian forecasting or artificial intelligence.

The benefits of using dosing software when compared with dosing nomograms or standard TDM processes include simplification of the process of calculating complex PK-PD parameters (such as AUC/MIC ratios). These PK-PD indices are being increasingly used in clinical practice to determine appropriate antimicrobial exposure targets. Measurement of surrogate targets, such as trough concentrations, may be alternatives in the absence of dosing software. However, as evidenced by vancomycin, target trough concentrations inconsistently correlate with the actual AUC/MIC targets for clinical efficacy, thereby risking drug toxicity [90].

Below is a brief description of the different approaches currently available for dosing software programs.

### 4.1. Linear Regression Based Dosing Software

A simpler form of dosing software is based on linear regression principles whereby two plasma concentrations at different time points are obtained and an algorithm calculates a drug clearance rate. This can then be used to calculate a dose adjustment regimen. Aminoglycoside dose optimisation with the program Aminoglycoside Levels and Daily Dose Indicator (ALADDIN) is one such example of this [91,92]. Outside of the ICU, the use of this program has been shown to produce dosing recommendations that are different from dosing nomograms but the predictive performance in terms of PK-PD target attainment have yet to be quantified. Interestingly, the use of ALADDIN may be associated with a lower incidence of nephrotoxicity compared to dosing with nomograms [92]. This potentially suggests linear regression dosing software may be superior to simpler methods such as dosing nomograms for dosing aminoglycosides, but this has yet to be prospectively examined in the ICU cohort.

As linear regression programs do not include population PK models, multiple samples of measured plasma concentrations are required in order to describe an individual PK profile [65]. These programs are unable to generate empiric starting doses based on patient covariates and do not consider any other patient data in their analysis and recommendations. As each set of concentrations are analysed individually, these programs do not have the ability to consider previous TDM results for the patient to further refine the accuracy of dosing recommendations, which may be necessary in the event of changes in a patient’s clinical condition. Collectively, this may impair a clinician’s ability to make the most accurate dosing recommendations in a timely fashion.

### 4.2. Population PK-Based Dosing Software

Dosing software programs that utilise population PK or statistical models could be considered advanced dosing nomograms. Patient covariates, or measured drug concentrations, are entered into the software and the program generates a starting dose recommendation or dosage adjustment, that aims to achieve a predefined antimicrobial exposure [93]. Unlike linear regression-based programs, population PK based models are effective with even one plasma concentration measurement as the program is able to utilise the underlying model to predict the necessary dose alterations needed to achieve a PK-PD target [92]. Accuracy typically improves if a second (or third) plasma concentration is included.

Thus far, population PK-based dosing software has shown some promise. For example, the program DoseCalc produced similar dosing recommendations to clinicians who performed complex manual calculations for aminoglycosides dosing when targeting an AUC_0–24_/MIC target [93]. The software also generated recommendations that were closer to manually calculated doses compared to a dosing nomogram when used in patients with renal dysfunction. Unfortunately, predictive performance using measured plasma concentrations was not evaluated.

One limitation associated with using population PK-based methods is the inability to use measured antimicrobial concentrations from the patient to further individualise the model (i.e., *a posteriori* PK parameter estimates) [94,95]. This may reduce the potential predictive accuracy of the program when new dosing recommendations are required in the event there is a change in a patient’s clinical parameters without obtaining further TDM concentrations. Hence, recommendations are extrapolated from the population PK model driving the software, not the individual patient’s PK parameters, which may potentially delay achievement of PK-PD targets. Additionally, many population PK model softwares do not contain models from critical care patients which may compromise accuracy when used in the critical care cohort [5].

### 4.3. Bayesian Forecasting Dosing Software

Software that utilises Bayesian forecasting typically uses population PK data to estimate a recommended dose that is likely to help achieve a predefined PK-PD target. However, Bayesian forecasting has the added benefit of using data (such as TDM results) to generate *a posteriori* PK parameter estimates that can strengthen and improve the accuracy of future dosing recommendations (Figure 3) [96]. They also have the ability to account for pathogen-specific parameters such as MIC where variations in the degree of susceptibility will alter the dosing regimens that are needed to achieve the PK-PD targets. In this way, Bayesian forecasting is able to account for any patient variations from the population that the initial PK model was built on. Similar to population PK based methods, Bayesian forecasting removes the need for multiple drug samplings as the program is able to utilise population PK data to predict likely antimicrobial concentrations and generate dosing requirements needed to achieve the PK-PD targets [97].

The use of Bayesian methods for optimising drug dosing was first described in the 1970s but required complex mathematical calculations that most clinicians were unlikely to be familiar with [98]. Combined with the lack of awareness then around the PK differences in patient populations and its impact on achieving antimicrobial PK-PD targets, this likely may have led to poor uptake amongst clinicians. It was not until advances with computer sciences that enabled dosing software to integrate Bayesian forecasting into the analysis were clinicians able to appreciate the potential benefits of using it to help optimise antimicrobial treatment in the critical care patient group. Examples of dosing software with Bayesian forecasting include but are not exclusive to Best Dose, ID-ODS, DoseMe, and TCI Works [5]. Some of these programs include population PK models for antimicrobials specific to the critically ill patients with sepsis built into the dosing software. Details of antimicrobial population PK models developed from critical care patient groups are provided in Appendix A.

To date, there are limited studies prospectively evaluating the performance of Bayesian dosing software in the ICU. In one prospective study involving critical care patients using cefepime, meropenem and piperacillin-tazobactam, the use of the dosing software ID-ODS resulted in 98% of patients achieving the predefined PK-PD target of time above MIC of the dosing interval (50% for piperacillin, 40% for meropenem and 60% for cefepime) [99]. Though there were no control groups to compare target attainment in the absence of dosing software, 22% of dosing software optimised patients received a different dose to standard doses that would have been used in the study ICU, which raises the possibility of inadequate or excessive antimicrobial exposure in those patients if standard doses were used, although MICs were generally low. Studies demonstrating positive clinical outcomes associated with attainment of PK-PD targets have been described in critically ill patients [10,33,34,100,101,102,103]. However, the use of Bayesian dosing software to achieve these targets and its impact on clinical outcomes have yet to be evaluated but are still needed.

### 4.4. Artificial Intelligence Software

A relatively new approach to optimising treatment of critically ill patients with sepsis in the ICU is with the use of computerised programs that utilise artificial intelligence (AI) in generating their recommendations. Unlike dosing software with Bayesian forecasting which uses PK and statistical modelling to individualise therapy to patients, AI software uses reinforcement learning to generate recommendations on appropriate interventions required to achieve predetermined targets for patients [104]. Artificial intelligence software examines data from large patient population databases to identify interventions associated with the target outcome and combines this information with an individual patient’s characteristics to determine the most appropriate intervention that will maximise the probability of achieving the predefined outcome [105,106]. If the predetermined outcome is not achieved, clinicians are able to relay this information back to the AI software which will then further refine its algorithm to alter recommendations to help achieve the target (similar to a trial-and-error approach).

Although there is preliminary evidence that highlights the potential utility of AI software for optimising the treatment of patients in the ICU [107], evidence for AI software use for optimising the dosing of antimicrobials in the ICU is lacking. The dosing software InsightRx, which utilises both Bayesian forecasting and AI [108], has been observed to accurately predict the AUC of vancomycin based on one trough concentration (AUC_InsightRx_: AUC_reference_ of 0.84 with 25th–75th median percentile of 0.77–0.88, accuracy was improved when two concentrations were used) [97]. In this study, InsightRx performed similarly well to other dosing software that utilise only Bayesian forecasting for predicting vancomycin AUC. InsightRx’s ability to recommend dosing changes and its impact on achieving target AUC has not been evaluated in critically ill patients.

One potential benefit of AI-based dosing programs over other dosing software is the potential for AI software to consider the impact of interacting drugs on antimicrobial concentrations (such as that mediated by cytochrome enzyme induction or inhibition). If embedded within the electronic health records, AI programs may potentially identify medications through its databases where concurrent prescribing with an antimicrobial result in alterations to antimicrobial plasma concentrations [109,110]. The software could potentially make proactive dosing recommendations to compensate for such drug interactions and maximise the likelihood that antimicrobial PK-PD targets could be achieved. However, this has yet to be prospectively evaluated in critically ill patients.

### 4.5. Challenges ahead for Widespread Dosing Software

As described above, there are currently many types of dosing software that utilise different models when generating dosing recommendations. This may pose a barrier to clinicians as there is insufficient data to support superiority of one model over another. Although complex programs that utilise Bayesian forecasting and/or AI may produce dosing recommendations that are more likely to achieve PK-PD targets, this has yet to be adequately examined in the critically ill cohort. Studies powered to examine the comparative predictive accuracy of dosing recommendations of these programs are needed before clinicians are able to confidently adopt one program over another into their clinical workflows. Additionally, studies powered to quantify clinical outcomes such as sepsis-related mortality are needed before widespread use can be recommended.

Another potential barrier associated with using dosing software in the clinical setting is the adequate knowledge and training required by users [111]. This includes familiarisation with the software as well as ensuring that they have sufficient understanding of antimicrobial PK-PD. Clinicians would need to develop knowledge of concepts related to MICs and the interrelationship with antimicrobial PK-PD so that appropriate exposure targets are selected. This may be lacking unless clinicians are trained in the areas of microbiology or infectious diseases [112]. Potential solutions may include utilisation and integration of specialised clinical pharmacists or pharmacologists into the critical care team who are trained in using dosing software to assist with complex dosing [113]. Furthermore, not all ICUs have access to the different drug assays needed to generate TDM results for antimicrobials which are required by software to generate optimised dosing recommendations.

Currently, most dosing softwares are either web-based or standalone applications [5,108]. This potentially detracts from the usability of the dosing programs given clinicians will be required to manually extract data from hospital-based medical records, input data into the dosing program, generate the recommended doses and then return back to the medical records to amend the dose [108]. In health services where clinicians are often time-poor, this may pose a significant barrier to wide-spread adoption of this technology. Several dosing software programs have been developed to integrate with hospital-based medical records for non-antimicrobial drugs outside of the ICU and these programs have been deemed satisfactory as perceived by clinicians [114]. Pleasingly, some commercial program developers have highlighted the ability for their dosing software for antimicrobials to be integrated with local electronic health record platforms, thus potentially improving clinical workflows in the ICU.

If research shows that dosing software confers patient-centred outcome benefits, then before being able to be widely used clinically, several challenges must be addressed. Currently, few dosing software programs are registered as medical devices with national regulatory bodies. Although clinicians are still responsible for accepting the recommendations from the dosing software program, a reliance on this technology for dosing is likely to raise concerns regarding the accuracy and safety of these programs if they are unregulated beyond the internal quality control processes of the software developers [115]. In the future, software developers may be required to register their dosing software with relevant regulatory bodies before health services are able to incorporate this technology into their workforce.

Although favourable cost outcomes from using dosing software have been reported [116,117], this analysis has yet to be conducted in critically ill patients. Costs that healthcare networks need to consider include the resources needed to train clinicians to use the software, costs associated with integrating dosing software with local electronic health records as well as the physical infrastructure needed to house dosing software along with the large amount of data that is likely to be generated from performing dosing simulations. A potential cost-efficient option for these networks to consider when investing in dosing software is to select software capable of providing support across a wide range of clinical settings, as opposed to programs that only specialise in one or two clinical areas.

## 5. Conclusions

Dosing of antimicrobials in critically ill patients with sepsis remains a challenging area. Due to the changes in physiology typically seen in these patients, adopting dosing regimens from non-critical care patients are unlikely to achieve desired PK-PD targets that are associated with optimal outcomes. There are several approaches that can be employed by clinicians to increase the likelihood of achieving these antimicrobial targets in critically ill patients. Where available, clinicians should consider using dosing nomograms and/or TDM to support their dosing strategies to improve the likelihood of achieving PK-PD targets that are associated with positive clinical outcomes. Dosing nomograms are generally easy to integrate into clinical practice and do not require extensive resources beyond clinician training. TDM can help identify patients who have not achieved PK-PD targets or who may have developed toxic concentrations which are predisposed to adverse effects. Clinicians can then use TDM results to alter dosing regimens by a magnitude they anticipate will achieve PK-PD targets or in conjunction with other tools such as dosing nomograms or dosing software programs to improve the dosing accuracy of these tools.

Other dose optimisation strategies that may have increased precision for achieving antimicrobial PK-PD targets, of which dosing software is a promising tool. Other dose optimisation strategies exist, such as dosing software, which may offer recommendations that achieve PK-PD targets with increased precision. Simpler dosing programs that utilise linear regression or population-based models may be suitable and are easier to implement, especially in centres that have limited resources. In centres with access to clinicians trained to alter antimicrobial dosages to achieve PK-PD targets, dosing software with Bayesian forecasting and/or AI may provide additional precision to improve antimicrobial dosing.

Importantly, future studies describing the clinical outcomes and cost-benefits associated with using dosing software and AI are needed and it is hoped that this data will help consolidate the utilisation of this technology in patients that are high risk of dying from infections. In the meantime, clinicians should at least consider using dosing nomograms and/or TDM to support their dosing strategies to improve the likelihood of achieving PK-PD targets that are associated with positive clinical outcomes.

## Figures and Tables

**Figure 1 pharmaceutics-12-00638-f001:**
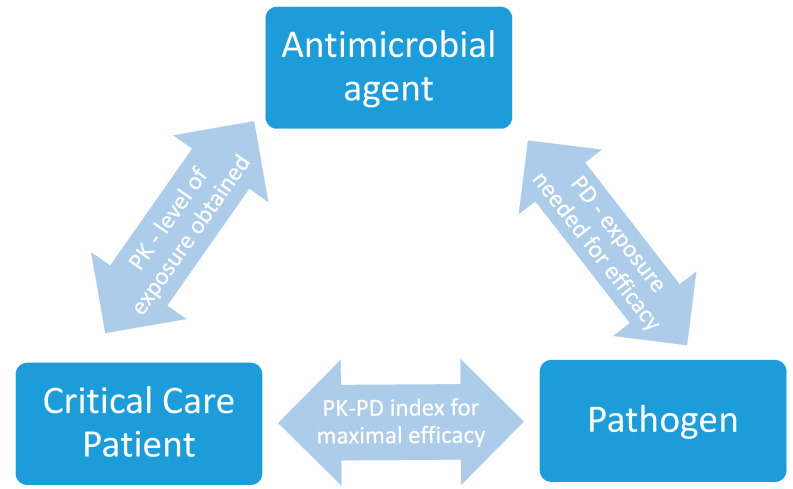
Dynamic interplay between the critical care patient, antimicrobial agent of choice, and the pathogen.

**Figure 2 pharmaceutics-12-00638-f002:**
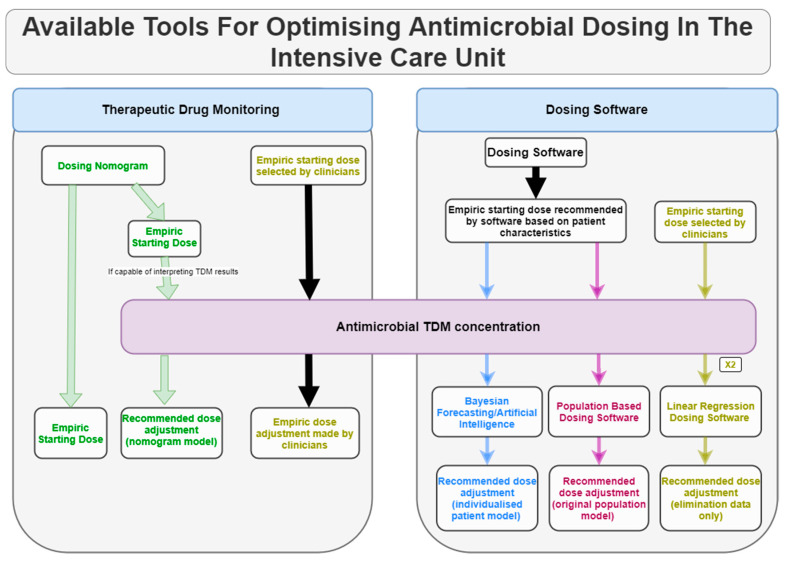
Range of available tools to assist clinicians with optimising the dosing of antimicrobials in the intensive care unit (ICU). Simple applications such as dosing nomograms and therapeutic drug monitoring (TDM) provide clinicians with basic tools needed to improve probability of achieving PK-PD targets and determine if they have been successfully achieved. Computerised programs such as dosing software (and some nomograms) are able to utilise antimicrobial TDM results to generate refined recommendations based on the model underpinning the software.

**Figure 3 pharmaceutics-12-00638-f003:**
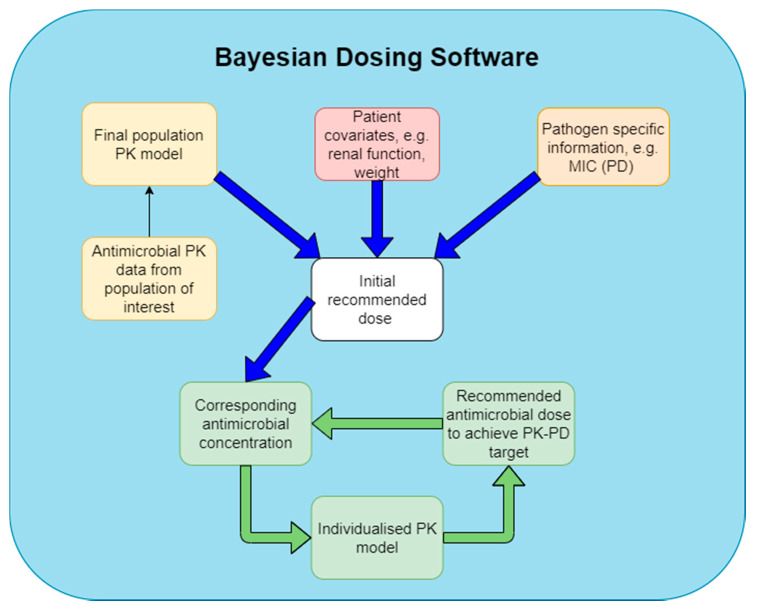
Bayesian forecasting based dosing software (blue) initially utilises patient covariates (red), PK properties of the antimicrobial (yellow), and pathogen PD properties (orange) to generate an initial recommended dose that is likely to achieve the PK-PD target associated with maximum efficacy. Resulting plasma exposure concentrations from this dose can then be used to produce a refined PK model that is individualised to a specific patient. A new optimised dose is then produced that is specific to the patient and can be further refined if needed with additional TDM data (green cycle).

**Table 1 pharmaceutics-12-00638-t001:** List of commonly used antimicrobial classes and their pharmacokinetic-pharmacodynamic (PK-PD) indices.

Class	PK-PD Index	Reference
Aminoglycosides	Cmax/MIC AUC_0–24_/MIC	[40]
Beta-Lactams	*f*T > MIC	[41]
Fluoroquinolones	Cmax/MIC AUC_0–24_/MIC	[33,42]
Glycopeptides	AUC_0–24_/MIC	[43]
Glycylcyclines	AUC_0–24_/MIC	[44,45]
Lincosamides	*f*T > MIC	[46]
Lipopeptides	Cmax/MIC AUC_0–24_/MIC	[47]
Macrolides	*f*T > MIC AUC_0–24_/MIC (azithromycin)	[48,49]
Oxazolidinones	*f*T > MIC AUC_0–24_/MIC	[50]
Polymyxins	AUC_0–24_/MIC	[51]
Triazoles antifungals	AUC_0–24_/MIC	[52]
Echinocandins	AUC_0–24_/MIC Cmax/MIC	[53]
Polyenes	Cmax/MIC	[54]

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
