# Peer review of "What Are the Current Approaches to Optimising Antimicrobial Dosing in the Intensive Care Unit?"

_pharmaceutics, 2020, doi:10.3390/pharmaceutics12070638_

Round 1

Reviewer 1 Report

The information provided by Chai et al. deals with the current approaches available to optimize antimicrobial dosing in intensive care. The “one-size-fits-all” approach that is commonly applied in many situations is innefective and inadequated to many patients. This study is therefore pertinent and, in light of the pandemic that the world is facing and the issues that are raised when it comes to antimicrobial infections, completely necessary. The authors reviewed the use of dosing monogram systems, therapeutic drug monitoring and even explored software-assisted approaches. The writting is clear, however the organization should be improved, namely the formating of the subsections. For instance, in section 4 it is not clear the difference between the bold entitled sections and the italic entitled sections. The italic section appears as if part of the bold section prior and it is not, a numering system may be important here.The mansucript is scientifically sound and the information reads very nicelly. However, it would be important for the authors to explore in further details what should be done in the future to overcome the limitations of the current dosing strategies and what should be the posture and the measures taken by hospital personnel when it comes to this decision. This information should be added to the conclusions section. They refer in the abstract that new studies should be done to implement new strategies to quantify the clinical and cost benefits associated with the approaches used to monitor/decide antimicrobial dosing; however, the authors did not propose alternatives. It would be important to refer some options in the conclusions section, namely future work. Finally, please reduce the number of keywords and change the lable of the tables for the top and not the bottom (that is used for figures). 

Author Response

Reviewer 1

  1. section 4 it is not clear the difference between the bold entitled sections and the italic entitled sections

Thank you for highlighting this. We have amended this to reflect that challenges associated with dosing software use are applicable to all dosing programs (page 13, line 386 of updated manuscript).

  1. However, it would be important for the authors to explore in further details what should be done in the future to overcome the limitations of the current dosing strategies and what should be the posture and the measures taken by hospital personnel when it comes to this decision. This information should be added to the conclusions section.

We have expanded our conclusion section to specifically mention the tools described in the review as suitable alternatives to current dosing strategies. We have also made suggestions on which tools can be adopted by clinicians based on the resources available to each ICU (page 14, line 451 of the updated manuscript).

  1. They refer in the abstract that new studies should be done to implement new strategies to quantify the clinical and cost benefits associated with the approaches used to monitor/decide antimicrobial dosing; however, the authors did not propose alternatives.

We have clarified in the abstract that the limitations in quantifying clinical outcomes are specific to dosing software and clinicians may consider using TDM and dosing nomograms which are already commonly used in ICUs as standard of care (page 1,line 20-25 of the updated manuscript).

  1. It would be important to refer some options in the conclusions section, namely future work. Finally, please reduce the number of keywords and change the lable of the tables for the top and not the bottom

As described in point 2, we have amended our conclusion to discuss this in page 14, line 451 of the updated manuscript. We have also removed keywords that appear in our review title and amended the label of the table.

Reviewer 2 Report

The authors present a narrative review regarding the current approaches to optimize antimicrobial dosing in ICU. In an era that medicine and clinical practice changes and shifts towards personalized and stratified approaches in clinical practice, the thematic is interesting -especially as stated in the abstract. However, although the authors show expertise from previous works, it gives a feeling of shallow approach without delving into the key points and does not present in details current state-of-the-art approaches to answer the question in the title. For example, what happens around the world regarding this issue? USA? Europe? Asia? Australia? What is missing and what should be added soon? How the barriers overcame in other drug categories in ICU? Overall the manuscript should be re-edited in order to be more precise and better address the several issues it deals with. Some additional comments follow: 

Abstract: What is the purpose of the review? Authors present the background (in a very good way) but not clearly define what this review aims to present. Such as in lines 116-122.

Introduction 29-33 some additional comments regarding sepsis/mechanisms/outcome/ treatment strategies would assist to introduce the theme better. But is sepsis the issue or generally the selection of the most appropriate antimicrobial agent in ICU? Generally a better description for antimicrobials is needed

Antibiotic resistance? A huge problem in ICU, where is in the text? Also, critically ill patients in ICU are under multiple therapies and therefore at a high risk for potential drug interactions. What is the impact on antimicrobial therapy of these issues? What solutions are there (from the systems presented etc.)

the sub-section of altered PK is not so necessary since it's not followed by any other sub-section in introduction.

Lines 39-41 it would be optimum to add some references to support this statement

Line 93: only total bound may be measured? Beer J, Wagner CC, Zeitlinger M. Protein binding of antimicrobials: methods for quantification and for investigation of its impact on bacterial killing. AAPS J. 2009;11(1):1‐12. doi:10.1208/s12248-008-9072-1

Regarding the f (unbound concentration) is it modulated in critically ill patients, thus a PK parameter that alters the PD effect? It is not so clear as it is presented, some edit/comment would be helpful.

Lines 99-100 Can the authors present these values between the two patient groups? It would add to their work (even with the potential miscalculations that said in lines 111-113).

Line 195-230. This is commented earlier (altered PK) and it would be optimal to cross-reference and comment it (unbound drug concentration etc.). It describes similar issues

Are there any techniques and bio-analytical methods for determination of antimicrobial concentration in the site of action i.e. lungs, Cerebrospinal Fluid etc. that could assist in improved estimation of therapy outcome? If yes why not added and commented?

The authors also avoid describing differences related with demographics (age, BMI etc.) and the presence of comorbidities and potential drug interactions etc. Why? It would add to their work the inclusion of such studies for ICU patients.

Line 210 please edit the parentheses

Line 237 computerised is appropriate term?

Line 236. The harnessing of modeling and simulation generates new approaches for optimum dosing scheme. Moreover, today the utilization of Pharmacometrics, PBPK/PD provides tools to improve nomograms, TDM approaches etc. Why the authors do not include them in their review? Reference 87 is 23 years back but since then, the advancements in M&S allows the generation of population-based PK/PD models for drugs (and for antimicrobials). Especially for PBPK models and the "bottom-up or top-down" data from "what if" scenarios (special population groups, drug interactions, optimum dosing etc.) are continuously applied and accepted from regulatory for several drug categories. Even if they are not applied in clinical level (in ICU for example) some comments would assist the manuscript to present the state-of-the-art approaches in modern healthcare. Examples:

  • Trivedi A, Lee RE, Meibohm B. Applications of pharmacometrics in the clinical development and pharmacotherapy of anti-infectives. Expert Rev Clin Pharmacol. 2013;6(2):159‐170. doi:10.1586/ecp.13.6
  • Rathi C, Lee RE, Meibohm B. Translational PK/PD of anti-infective therapeutics. Drug Discov Today Technol. 2016;21-22:41‐49. doi:10.1016/j.ddtec.2016.08.004
  • Barrett JS, Della Casa Alberighi O, Läer S, Meibohm B. Physiologically based pharmacokinetic (PBPK) modeling in children. Clin Pharmacol Ther. 2012;92(1):40‐49. doi:10.1038/clpt.2012.64
  • Lu J, Goldsmith MR, Grulke CM, et al. Developing a Physiologically-Based Pharmacokinetic Model Knowledgebase in Support of Provisional Model Construction. PLoS Comput Biol. 2016;12(2):e1004495. Published 2016 Feb 12. doi:10.1371/journal.pcbi.1004495
  • Gaohua L, Wedagedera J, Small BG, et al. Development of a Multicompartment Permeability-Limited Lung PBPK Model and Its Application in Predicting Pulmonary Pharmacokinetics of Antituberculosis Drugs. CPT Pharmacometrics Syst Pharmacol. 2015;4(10):605‐613. doi:10.1002/psp4.12034
  • Thémans P, Marquet P, Winkin JJ, Musuamba FT. Towards a Generic Tool for Prediction of Meropenem Systemic and Infection-Site Exposure: A Physiologically Based Pharmacokinetic Model for Adult Patients with Pneumonia. Drugs R D. 2019;19(2):177‐189. doi:10.1007/s40268-019-0268-x

Lines 337-348 evaluating the performance of Bayesian dosing software in the ICU. Under what terms according to authors opinion this should be achieved?

Lines 347-348 please edit not clear meaning.

Line 353. Are there any studies regarding clinicians’ awareness for using these approaches? The role of clinical pharmacist is indisputable, but it would be better if some relative references would be added.

Line 358. The authors repeat the difficulties for application of validated bio-analytical methods in TDM for some clinics so there is the question, what happens in this case? nomograms? Any proposals from other works or this one would add value to the manuscript.

Lines 361-368. Usually, developed software are evaluated regarding interoperability between several systems. In what extend the dosing software are interoperable with EHR/EMRs? Any examples? In addition, Lines 370-378 what the guidance describes? some comments would be helpful -especially of what should be added.

Lines 752. The appendix is helpful, but it would be better commented in the main text. There are lot of studies here that the review could analyze further.

Author Response

Reviewer 2

  1. For example, what happens around the world regarding this issue? USA? Europe? Asia? Australia? What is missing and what should be added soon? How the barriers overcame in other drug categories in ICU? section 4 it is not clear the difference between the bold entitled sections and the italic entitled sections

Thank you for this comment and please allow us to clarify. We have described in our article that there are various differences to TDM practices across countries and institutions that may impact on achieving antimicrobial targets (page 7, line 193-195 of revised manuscript). We feel that attempting to include information regarding breakdown of practices in each country or institution is outside the scope of this invited article and indeed, we think that practice is so heterogeneous within and between countries, that it is too complex an issue to tackle as well. Our approach was to provide the various options that are available for use and allow the reader to choose that approach which may be most feasible for local use.

As this article specifically focuses on the challenges associated with antimicrobial dosing, we have not explored the impact of PK-PD changes and potential solutions on drug for non-infection indications.

  1. What is the purpose of the review? Authors present the background (in a very good way) but not clearly define what this review aims to present. Such as in lines 116-122..

Thank you for highlighting this. We have now specified the aim more clearly in the abstract. The purpose of this review was also stated in the main text of the introduction in page 4, lines 124-125 of the revised manuscript.

  1. the sub-section of altered PK is not so necessary since it's not followed by any other sub-section in introduction.

The altered PK section is a subsection of the introduction. We have included this as it serves to separate the introduction into several areas by topic (PK vs PD) to improve readability by the audience. A sentence has been added immediately preceding this section to more clearly articulate the value of this information to the reader.

  1. Introduction 29-33 some additional comments regarding sepsis/mechanisms/outcome/ treatment strategies would assist to introduce the theme better. But is sepsis the issue or generally the selection of the most appropriate antimicrobial agent in ICU? Generally a better description for antimicrobials is needed.

Thank you for this comment. We agree that selecting the right antimicrobial agent is critical in treatment success and have highlighted this in the introduction (page 1, line 30 of the updated manuscript) Alterations in PK-PD is another important factor to consider as it affects the achievement of target antimicrobial concentrations and is the focus of our review. We have now expanded the introduction to specifically mention the physiological factors and interventions administered to this cohort of patients are responsible for affecting this important parameter of antimicrobial therapy (page 1, lines 34-36 and page 2, lines 52-3, 58-61 of the updated manuscript). The mechanisms are further elucidated in the PK and PD sections of the introduction.

  1. Antibiotic resistance? A huge problem in ICU, where is in the text? Also, critically ill patients in ICU are under multiple therapies and therefore at a high risk for potential drug interactions. What is the impact on antimicrobial therapy of these issues? What solutions are there (from the systems presented etc.)

Thank you for this important comment. We agree that these are very important considerations in patients in the ICU. The increased presence of resistant pathogens are mentioned in the abstract (page 1, line 15 of revised manuscript), in the PD section (page 4, line 103) and the impact on resistance (page 4, lines 107-8). Potential options to account for pathogens with higher MIC are describe in lines 319-21 in the Bayesian dosing section. We appreciate that drug interactions are very important considerations for clinicians to consider when selecting the appropriate antimicrobial agent and their dosages. Current dosing programs are unable to predict this effect on antimicrobial concentrations. We have however, highlighted the potential for artificial intelligence to consider this, but requires future evaluation (page 13, lines 385-392 of the revised manuscript). 

  1. Lines 39-41 it would be optimum to add some references to support this statement

Thank you for highlighting this. We have now referenced this statement (page 2, lines 42, 44 of the updated manuscript)

  1. Line 93: only total bound may be measured? Beer J, Wagner CC, Zeitlinger M. Protein binding of antimicrobials: methods for quantification and for investigation of its impact on bacterial killing. AAPS J. 2009;11(1):1 doi:10.1208/s12248-008-9072-1

Thank you for this comment and please allow us to clarify. Whilst total concentrations could broadly be considered acceptable for low protein bound drugs, when establishing a dose optimisation program, total concentrations can not be reliably inferred for moderately-to-highly protein bound drugs and hence an unbound measurement approach is required. Measurement of bound concentrations would not be helpful. To further clarify this perspective, we have provided a reference reporting assaying of free antimicrobial concentrations in the TDM section (page 8, lines 241-244 of the updated manuscript).

  1. Regarding the f (unbound concentration) is it modulated in critically ill patients, thus a PK parameter that alters the PD effect? It is not so clear as it is presented, some edit/comment would be helpful.

Thank you and we have changed page 2, line 61-4 (and pharmacodynamic section lines 87-99) of the updated manuscript to explicitly mention a reduction in plasma albumin concentrations is common in critically ill patients and leads to altered drug concentrations.

  1. Lines 99-100 Can the authors present these values between the two patient groups? It would add to their work (even with the potential miscalculations that said in lines 111-113).

We would request not to include the actual values as these are observational data encompassing a large amount of pathogens across several countries. Doing so would distract the reader the main messages of our review. We hope this is acceptable to the Editor

  1. Line 195-230. This is commented earlier (altered PK) and it would be optimal to cross-reference and comment it (unbound drug concentration etc.). It describes similar issues

Thank you and although we agree with the Reviewer’s suggestions. We have highlighted the important issues of unbound concentrations as suggested linking the two sections. This also is exemplified by use of the ‘f’ symbol denoting free drug values as shown throughout lines 87-99.

  1. Are there any techniques and bio-analytical methods for determination of antimicrobial concentration in the site of action i.e. lungs, Cerebrospinal Fluid etc. that could assist in improved estimation of therapy outcome? If yes why not added and commented?

Thank you for this important comment. We have described in page 7, line 216-219 of the updated manuscript that the challenges with measuring antimicrobial concentration in these sites relates to the invasiveness of the sampling technique, not the actual bioanalytical method. We have also included a new sentence (page 7, line 205-207) highlighting limitations with current models in extrapolating concentrations at the site of infection from PK in blood. This is further expanded on in page 8, lines 225 describing the need for more accurate data if we are to extrapolate from plasma concentrations and to also consider the impact on patient outcomes.

  1. The authors also avoid describing differences related with demographics (age, BMI etc.) and the presence of comorbidities and potential drug interactions etc. Why? It would add to their work the inclusion of such studies for ICU patients.

Thank you and please allow us to clarify. We acknowledge that these are important factors to consider with dosing of some antimicrobials, however, we would contend that these issues are not unique to critically ill patients. Hence we have elected not to focus on these characteristics in our review that primarily focusses on challenges associated with critically ill patients and potential solutions specific to this patient cohort.

  1. Line 210 please edit the parentheses

We have removed them as requested.

  1. Line 237 computerised is appropriate term?

We have rewritten this sentence to better reflect appropriate terminology (line 245 of the updated draft)

  1. Line 236. The harnessing of modeling and simulation generates new approaches for optimum dosing scheme. Moreover, today the utilization of Pharmacometrics, PBPK/PD provides tools to improve nomograms, TDM approaches etc. Why the authors do not include them in their review? Reference 87 is 23 years back but since then, the advancements in M&S allows the generation of population-based PK/PD models for drugs (and for antimicrobials). Especially for PBPK models and the "bottom-up or top-down" data from "what if" scenarios (special population groups, drug interactions, optimum dosing etc.) are continuously applied and accepted from regulatory for several drug categories. Even if they are not applied in clinical level (in ICU for example) some comments would assist the manuscript to present the state-of-the-art approaches in modern healthcare. Examples:

In regards to reference 23, we acknowledge this is an old paper, but it discusses important aspects on the use of a dosing program with linear regression techniques (such as ALADDIN) that is still used in some clinical areas.

We elected not to focus on pharmacometrics as it is predominantly concerned with the development process of antimicrobial dosing regimens, including those models that are included within dose software, and as such is not unique to our topic, optimising the dosing of antimicrobials in critically ill patients. We acknowledge there are elements of the pharmacometrics topic that are important to this review (such as PK-PD changes in critically ill patients, optimising therapy by targeting PK-PD index) and as such we have been discussed this in the body of the review. We feel that this is sufficient as it provides the necessary context to understand the importance of the interventions we are discussing (such as the use of dosing nomograms/TDM/dosing software) without providing additional information that may detract from this. 

  1. Lines 337-348 evaluating the performance of Bayesian dosing software in the ICU. Under what terms according to authors opinion this should be achieved?

We have discussed this in page 12, lines 358-359 of the updated manuscript.

  1. Lines 347-348 please edit not clear meaning.

Thank you for highlighting this, we have rewritten this sentence to improve readability (line 358-359 of the updated manuscript)

  1. Line 353. Are there any studies regarding clinicians’ awareness for using these approaches? The role of clinical pharmacist is indisputable, but it would be better if some relative references would be added.

This sentence aims to discuss the challenges associated with using dosing software. One challenge discussed is the lack of awareness by general clinicians regarding usage of PK-PD targets if dosing software is to be used. We have now provided a reference that describes this lack of understanding by non-microbiological/infectious diseases clinicians. (page 13, line 409-410 of the updated draft)

  1. Line 358. The authors repeat the difficulties for application of validated bio-analytical methods in TDM for some clinics so there is the question, what happens in this case? nomograms? Any proposals from other works or this one would add value to the manuscript.

Thank you for highlighting this. This was also pointed out by reviewer 1 requesting clarification on the approaches to optimising antimicrobial therapy given the limitations of each tool. We have now made clearer suggestions of our recommended approach in the conclusion as requested by reviewer 1 (see page 14, lines 451 of the updated manuscript).

Lines 361-368. Usually, developed software are evaluated regarding interoperability between several systems. In what extend the dosing software are interoperable with EHR/EMRs? Any examples? In addition, Lines 370-378 what the guidance describes? some comments would be helpful -especially of what should be added.

We have now included an example of improved usability associated with EHR integration and highlighted the ability of some software developers to integrate dosing software with local EHR. (page 14, line 421-426 of the updated draft).

Thank you for this comment. In regards to line 370-378 (first submission), we have discussed how dosing programs are likely to be required to be registered with regulatory bodies before widespread use is likely to be accepted (page 14, in line 429-436 of the updated manuscript). A reference for this is in reference 112 of the updated draft.

  1. Lines 752. The appendix is helpful, but it would be better commented in the main text. There are lot of studies here that the review could analyze further.

Thank you for this suggestion. We initially considered this but elected to place it in the appendix as it would reduce the bulk of the body of the text as the table is five pages long.  By being included as an appendix, it is still available for the reader to access and analyse individually depending on the topic of their interest.

Reviewer 3 Report

This is an interesting review of possible approaches for optimization of therapy in ICUs. It’s a very important topic. Authors presented many methods yet I believe they lack an important angle. Although they presented Bayesian forecasting they don’t mention any artificial intelligence / machine learning techniques (AI/ML). Today even Bayesian methods are used as hybrids with AI/ML. However not only Bayesian methods are at scope here. An additional buzzword here would be “explainable AI”.

I believe Authors should add a chapter of AI/Ml at the end of their paper to provide complete picture. I am aware that many AI/ML solutions are still in the development phase (i.e. DeepAISE) yet there are some examples of at least preliminary implementations as well (Artificial Intelligence Clinician, InsightRX).

Author Response

Reviewer 3

  1. In response to the request for inclusion of artificial intelligence (AI).

Thank you for your suggestion regarding including AI. We have included a new subsection under dosing software (4.4) that specifically discusses the role of AI in optimising the dosing of antimicrobials (line 362-92 of the updated manuscript). We note that the majority of evidence around the use of AI in the ICU is around the prescription of certain treatments (such as decision of when to commence fluids and vasopressors). There are no antimicrobial dosing software programs that are marketed to only use AI in their algorithm. However, we have included information on InsightRx to provide a balanced analysis of tools that utilise AI to a certain degree. 

Round 2

Reviewer 2 Report

The authors present an updated version of their manuscript addressing some of the issues of their first version. As the issue suggests, in the era of precision medicine, optimizing drug exposure (such as for antimicrobials) is an essential part of appropriate pharmacotherapy, as it may ultimately affect clinical outcomes. Thus there is stressful need to utilize state-of-the-art approaches which today shift medicine from “one-size-fits-all” towards personalized and stratified approaches (i.e. antibiotic stewardship programs). The comments made for the initial version were in the basis of suggestions to improve the manuscript and the soundness of the review. The manuscript can be processed further for publication and it is in discretion of the authors to address the comments made.

Line 69. The statement is not accurate. Premarketing studies at phase II (~200 participants) and phase III (~2-3000 participants, multicenter planned) are single or double blind clinical trials conducted in patient cohorts for PK evaluation and to test efficacy, effectiveness as well as side effects for a drug. Please edit.

 Although the authors recognize the importance issues related with demographics (age, BMI etc.) the presence of comorbidities and drug interactions in ICU patients their reluctance in delving into them subtracts much of the review’s soundness (see also Rev Esp Quimioter . 2019 Sep;32 Suppl 2(Suppl 2):42-46 and Emerg Med Clin North Am. 2017 Feb;35(1):199-217. doi: 10.1016/j.emc.2016.09.007.). A simple search in the literature results in studies that report up to 20-40% of the total contra-indicated combinations and major drug-drug interactions to be related with antimicrobials in ICU. Studies that explore morbidity and mortality due to DDIs in ICU patients (including reasons such as drug-antimicrobials interactions) are available. The authors in their reply provide the page 13 Lines 385-392 stating “Current dosing programs are unable to predict this effect on antimicrobial concentrations. We have however, highlighted the potential for artificial intelligence to consider this, but requires future evaluation (page 13, lines 385-392 of the revised manuscript).”

Quoting from these lines (that are in lines 498-502, probably due to differences between manuscript versions):

  • “One potential benefit of AI based dosing programs over other dosing software is the potential for AI software to consider the impact of other medications on antimicrobial concentrations (such as  cytochrome enzyme induction or inhibition). If embedded within the electronic health records, AI programs may potentially identify medications through its databases where concurrent prescribing with an antimicrobial result in alterations to antimicrobial plasma concentrations [108,109]”

Where exactly is highlighted the need that dosing programs that are unable to predict DDIs (or the importance of the field) and the potential of AI to consider this? The mention of CYP induction or inhibition can be considered as a highlight? Till deep-learning is applied in clinical level, the problem will continue to exist? The nomograms will fail and patients will die due to DDIs? Why the authors don’t suggest with one sentence that due to the importance of the problem, some solutions not only are available but also evaluated? (See Antibiotics 2020, 9, 19; doi:10.3390/antibiotics9010019)

The second issue of the manuscript is the avoidance regarding pharmacometrics. Yes, the authors are correct that PK-PD M&S is mostly applied in development processes, although nowadays it also used to update drug-labels for the clinical use of drugs. This was the reason for the comment regarding the “old” reference 87, now 91 in the revised manuscript. The comments of the initial review regarding PK -related problems of drugs (total bound, fu, fublood/futissue, AUC, MIC etc.), predictions of concentrations (free and bound) in the site of action (blood, lungs, CSF) dosing simulations in special population groups etc. have been investigated within pharmacometric approaches of M&S and moreover initiatives have been established from regulatory (FDA, EMA) for utilization of pharmacometrics in antibiotic research and minimization of resistance. It is odd authors’ reluctance to briefly present them as part of potential solution (even if their adoption in clinical setting fails).  See: Expert Rev Clin Pharmacol. 2013 Mar;6(2):159-70. doi: 10.1586/ecp.13.6.

Author Response

Reviewer 2

  1. Line 69. The statement is not accurate. Premarketing studies at phase II (~200 participants) and phase III (~2-3000 participants, multicenter planned) are single or double blind clinical trials conducted in patient cohorts for PK evaluation and to test efficacy, effectiveness as well as side effects for a drug. Please edit

Thank you for this comment. We have edited this sentence to remove the mention of clinical studies, but instead focus on PK data instead as the driver of the development of dosing regimens. This will reduce the need to explore the different phases of clinical trial development which is not the focus of our review article. See line 45-9 of page 2

  1. Where exactly is highlighted the need that dosing programs that are unable to predict DDIs (or the importance of the field) and the potential of AI to consider this? The mention of CYP induction or inhibition can be considered as a highlight? Till deep-learning is applied in clinical level, the problem will continue to exist? The nomograms will fail and patients will die due to DDIs? Why the authors don’t suggest with one sentence that due to the importance of the problem, some solutions not only are available but also evaluated? (See Antibiotics 2020, 9, 19; doi:10.3390/antibiotics9010019)

We agree that drug interactions are clinically important considerations that clinicians are required to be aware off when prescribing antimicrobials in and outside of the ICU. As our review focuses on tools that are currently available to clinicians, we are only able to provide evidence to clinicians based on what is reported in the literature. As previously described, no currently marketed dosing software program has the ability to consider the impact of drug interactions and suggest appropriate dosing alterations required to compensate for this effect. We have mentioned in the article (see line 388-390 page 13), the potential utility of AI. This includes the capacity for AI to consider interacting medications and potentially make suggestions that can compensate for this effect. This is certainly still in its infancy and requires further evaluation, but serves as a platform to spark research ideas that clinicians and software developers can consider for newer iterations for future dosing programs. However, this is a highlight complex area and fraught with uncertainty of the magnitude of effects of drug interactions on drug exposure and efficacy – we are not sure that ‘deep learning’ can address this, particularly if information about drug exposure is not being fed back into the system. It will be an interesting area to support and monitor in future years.   

With reference to the Antibiotics 2020 article, it certainly raises an important issue that the ability of different programs to detect drug interactions is dependent on the database driving the platform. The ability to for AI programs to detect interacting medications is already discussed in our article, see lines 394-6 in page 13 of our article. However, it still does not provide any additional guidance to our topic on how dosing software can compensate for these interactions and adjust the dosing of antimicrobials when exposed to interacting drugs. This requires further evaluation with future iterations of dosing programs, which is discussed in our article (see line 394-401 of page 13).

  1. The second issue of the manuscript is the avoidance regarding pharmacometrics. Yes, the authors are correct that PK-PD M&S is mostly applied in development processes, although nowadays it also used to update drug-labels for the clinical use of drugs. This was the reason for the comment regarding the “old” reference 87, now 91 in the revised manuscript. The comments of the initial review regarding PK -related problems of drugs (total bound, fu, fublood/futissue, AUC, MIC etc.), predictions of concentrations (free and bound) in the site of action (blood, lungs, CSF) dosing simulations in special population groups etc. have been investigated within pharmacometric approaches of M&S and moreover initiatives have been established from regulatory (FDA, EMA) for utilization of pharmacometrics in antibiotic research and minimization of resistance. It is odd authors’ reluctance to briefly present them as part of potential solution (even if their adoption in clinical setting fails).  See: Expert Rev Clin Pharmacol. 2013 Mar;6(2):159-70. doi: 10.1586/ecp.13.6.

Thank you for your comment, but we would say it is unfair to state that we are ‘reluctant’ to discuss? As the Reviewer would be aware, pharmacometrics and dosing simulation is a core part of dose-finding during drug development and comprises dosing algorithms in dosing software. As we state repeatedly in the paper, the pharmacokinetics of the patient population must be described so that effective doses can be identified. This is pharmacometrics and the process includes dosing simulations. We deliberately try to not to include too much jargon so as not to confuse readers that are more likely clinically-focused and less expert than the reviewer clearly is. We would point out that the comment about any dosing regimen currently licensed for ‘minimization of resistance’ is likely incorrect and we are unware of any such drug with a licensed dose specifically aimed at this endpoint. Regarding the Expert Rev Clin Pharmacol. 2013 paper, it is aimed at dosing regimen identification during drug development whilst our paper seeks to focus on clinical practice. Regarding relevant topics in this paper, we also include these in our paper as follows: 

  1. PK/PD concepts and how they affect dosing of antimicrobials. This is discussed is our article from lines 41 – 131 (page 2-4).
  2. PK-PD indices associated with efficacy. This is discussed in our article from lines 95 – 126 (pages 3-4)
  3. Optimising dosing regimen based on pathogen PD. This is discussed in our article from lines 328 – 330 in pages 10 and Figure 3.
  4. Regulatory bodies and their use of PK-PD modelling to assess antimicrobial efficacy. We have now included a section on the discussion of pharmacometrics in the drug approval process in the introduction (line 33-36 page 1).

Reviewer 3 Report

Great job. Congrats to the Authors! I'd suggest to put AI to keywords. It is a go

Author Response

Thank you for this suggestion. We have included artificial intelligence as a keyword.